# Fitts’ Tapping Task as a New Test for Cognition and Manual Dexterity in Multiple Sclerosis: Validation Study

**DOI:** 10.3390/medicina59010029

**Published:** 2022-12-23

**Authors:** Klaudia Duka Glavor, Bianca Weinstock-Guttman, Gorka Vuletić, Iva Vranić Ivanac, Nataša Šimić, Thomas J. Covey, Dejan Jakimovski

**Affiliations:** 1Department of Neurology, General Hospital Zadar, 23000 Zadar, Croatia; 2Department of Health Studies, University of Zadar, 23000 Zadar, Croatia; 3Jacobs MS Center, Department of Neurology, Jacobs School of Medicine and Biomedical Sciences, University at Buffalo, State University of New York, Buffalo, NY 14202, USA; 4Department of Psychology, Faculty of Humanities and Social Sciences, University J. J. Strossmayer in Osijek, 31000 Osijek, Croatia; 5Department of Psychology, University of Zadar, 23000 Zadar, Croatia; 6Division of Cognitive and Behavioral Neurosciences, Department of Neurology, Jacobs School of Medicine and Biomedical Sciences, University at Buffalo, State University of New York, Buffalo, NY 14214, USA; 7Buffalo Neuroimaging Analysis Center (BNAC), Department of Neurology, Jacobs School of Medicine and Biomedical Sciences, University at Buffalo, State University of New York, Buffalo, NY 14203, USA

**Keywords:** fitts tapping task, psychomotor performance, manual dexterity, cognitive processing speed, multiple sclerosis

## Abstract

*Introduction.* Studies suggest that people with multiple sclerosis (pwMS) experience continuous and subclinical physical worsening, even as early as their disease diagnosis. Validating sensitive and reproducible tests that can capture subclinical disease activity early in the disease are clinically useful and highly warranted. We aimed at validating the utility of Fitts’ Tapping Task (FTT) as reproducible measure of psychomotor performance in pwMS. *Materials and Methods.* Thirty newly-diagnosed pwMS (within 2 years of diagnosis and Expanded Disability Status Scale; EDSS ≤ 2.0), 30 people with migraine (pwMig), and 30 healthy controls (HCs) underwent a psychomotor assessment using the FTT, O’Connor hand dexterity test, and Visual Reaction Time Test (VRTT). Hand strength was measured using a hand-grip dynamometer. Subjects also provided patient-reported outcomes (PROs) using the 36-Item Short Form Survey (SF-36). Intrarater and interrater reproducibility was acquired on 5 HCs by two independent operators. Test–retest reproducibility was determined in 5 pwMS over a 1-week follow-up. Eight pwMS returned for the same test procedures 2 years after the baseline assessment. Bland–Altman plots were used to determine the minimally detectable change (MDC) and logistic regression models determined the ability to differentiate between newly-diagnosed pwMS and HCs. *Results.* FTT exhibited a high intrarater and interrater reproducibility (interclass correlation coefficient of 0.961, *p* < 0.001). The test–retest demonstrated an MDC of the average FTT at > 15%. PwMS had significantly a slower FTT time and O’Connor dexterity time when compared to pwMig and HCs (*p* < 0.001 for both). Higher Fitts’ difficulty levels (4th and 6th difficulty) and average performance on the O’Connor test were able to differentiate newly-diagnosed pwMS from HCs with 80% accuracy (*p* < 0.01). Slower FTT performance was correlated with worse PROs due to physical health. Over the 2-year follow-up, and despite being clinically stable (no change in EDSS), 6 out of 8 (75%) pwMS had more than a 15% worsening in their average FTT time. *Conclusions.* FTT is a highly-reproducible test for measuring psychomotor performance in newly-diagnosed pwMS. FTT can capture insidious worsening in psychomotor performance and cognitive function in early stages of MS.

## 1. Introduction

Multiple sclerosis (MS) is a chronic neuroinflammatory and neurodegenerative disease of the central nervous system (CNS) and is the most common reason for neurological disability in the working population [1]. In addition to the physical disability affecting the lower and upper extremities, a varying number of people with MS (pwMS) also exhibit cognitive impairment, particularly seen within the domain of information processing speed [2]. Based on the Multiple Sclerosis Outcome Assessment Consortium (MSOAC), a collaboration between multiple governmental, academic, and private MS partners, cognitive and hand impairments are considered as core outcomes in clinical MS care and as outcomes in regulatory trials. The current gold standards for measuring such impairments are the Symbol Digit Modalities Test (SDMT) and 9-hole peg test (9HPT), for cognitive and hand impairment, respectively [3,4]. Questions regarding the reliability, clinically relevant changes, and multiple proposed cut-off values for group and individual differences of the aforementioned tests are continuously discussed.

Based on this background, developing sensitive tools that can capture a previously undetectable slowing of hand dexterity and psychomotor performance in MS are of particular importance. For an example, a specially-engineered and sensor-equipped glove which quantifies the time of finger opposition has been repeatably shown to discriminate pwMS from HCs [5,6]. This glove can also measure subclinical hand impairment in otherwise asymptomatic radiologically isolated syndrome (RIS) cases [7]. In the later stages of the disease and in patients with already impaired lower extremity Fitts function, the monitoring and preservation of hand dexterity is critical to the patient’s life independence and overall well-being [8]. Moreover, determining the treatment efficacy in regulatory trials for more disabled pwMS would require a responsive test that quantifies psychomotor function and hand dexterity. As an example, a large phase-3 trial that investigated the effect of ocrelizumab in significantly disabled secondary-progressive and primary-progressive pwMS has already implemented 9HPT as the predetermined primary outcome (O’HAND trial, NCT04035005) [9]. These arguments suggest the need to develop, standardize, and validate sensitive tests that are easily-available and responsive to changes in hand dexterity/psychomotor performance.

Fitts’ law states that the time required for a person to tap (using a finger) or move a pointer to a target area is directly proportionate to the distance and inversely related to the size of the target. While this principle is heavily utilized in the design of software interfaces, the same law can be employed towards measuring human motor capacity [10]. The amount of information that the CNS needs to process before it initiates and determines the correct motor movement amplitude is commonly expressed through a measure called bits of uncertainty/difficulty. In the original work by Fitts and Peterson published in 1954 and 1964, Fitts’ Tapping Task (FTT) was used to examine the average motor capacity of healthy adults, and was estimated to be between 10 and 12 bits per second [11,12]. Only one previous work by Ternes et al. utilized the FTT in 22 pwMS and 22 healthy controls (HCs) and showed that such psychomotor analyses can provide useful and comprehensive information regarding patients’ upper-limb movement which included planning, visual control, and accuracy [13]. However, the FTT has only been studied in a heathy aging population and has not been systematically studied and validated in pwMS. While 9HPT and SDMT are validated hand dexterity and cognitive processing speed tests for assessment of pwMS, they measure motor and cognitive functions separately. Given that FTT is related to more complex psychomotor processes (represents the span of processes from perceptual, to premotor, to motor planning, to output), it may have greater relevance to real-world functions that are meaningful to pwMS.

Our study aimed at determining the reproducibility and construct validity of FTT in a matched cohort of pwMS, HCs, and negative controls (people with migraine (pwMig)). We hypothesized that FTT would serve as a good diagnostic differentiator between newly-diagnosed pwMS, pwMig, and HCs. Identifying early psychomotor impairments within newly-diagnosed pwMS may provide additional predictive value in determining future decline (ecological validity). The lack of significant motor involvement in pwMig may further help at differentiating the changes that drive FTT performance. Lastly, we hypothesized that FTT would have good reproducibility and its outcomes would correlate with patient-reported physical and cognitive limitations.

## 2. Materials and Methods

### 2.1. Study Groups

The study consisted of three equal groups of 30 newly-diagnosed pwMS, 30 pwMig, and 30 HCs (age-, sex- and years of education-matched at a 1:1:1 ratio) that were enrolled at the Neurology Department at the General Zadar Hospital and the Psychology Department of the Zadar University. The inclusion criteria for the pwMS were: (1) 18 to 65 years old; (2) to be diagnosed based on the 2010-revised McDonald criteria [14]; (3) to be clinically stable and on disease-modifying therapy (DMT) at the time of study enrollment; (4) to have an Expanded Disability Status Scale (EDSS) equal or lower than 2.0 [15]; and (5) diagnosed with MS within the last 2 years of study enrollment. The exclusion criteria for the pwMS were: (1) pregnant and nursing mothers; (2) having clinical relapse or the presence of radiological activity within 30 days of study enrollment; and (3) being prescribed any psychoactive medications that may influence the psychomotor performance (e.g., modafinil, benzodiazepines, amphetamines, and dextroamphetamines). The inclusion criteria for the pwMig were: (1) 18 to 65 years old; and (2) being diagnosed with migraine based on the second International Classification of Headache Disorders–ICHD-3 [16]. The pwMig had similar exclusion criteria: (1) pregnant and nursing mothers; (2) no acute migraine attack within 30 days; and (3) no use of psychoactive medications. Lastly, the HCs were included based on the same age criteria. The HCs were excluded based on: (1) having current or history of major neurological disorder; and (2) presence of major depressive disorder or on any psychoactive medications.

All study subjects were examined by a certified experienced neurologist. The disability of the pwMS was scored based on the EDSS scale. EDSS worsening was determined using the common phase-3 trial criteria where pwMS were classified as worsened if they had (1) an EDSS increase of ≥1.0 EDSS point when baseline EDSS was 1.0–5.0; or (2) an EDSS increase from a baseline EDSS of 0–2.0. Based on the clinical presentation and disease history, all pwMS were classified as relapsing-remitting MS (RRMS) [17]. The study was approved by the local Institutional Review Board (IRB) and all subjects provided informed consent using a written consent form.

For the purpose of the reproducibility analysis, additional groups of 5 pwMS and 5 HCs were recruited. The 5 pwMS were part of the test–retest analyses and performed the FTT both at index and short 7 day follow-up. The 5 HCs also performed the FTT, and their data were utilized as part of the intrarater and interrater reproducibility.

### 2.2. Psychomotor Assessment of the pwMS, pwMig, and HCs

The psychomotor performance of the three study groups was determined using a battery of psychomotor and hand dexterity tests. The FTT are a series of psychomotor tasks where the subjects are required to alternate hitting two identical strip-targets. Target widths were 0.5 cm, 1.0 cm, 2.0 cm, and 4.0 cm, and movement amplitudes were 4.0 cm, 8.0 cm, and 16.0 cm, respectively. The test is built upon Fitts’ law. Combinations between the strip-target widths and the amplitude distance between targets create 12 specific tasks that range in difficulty from 1 to 6 bits and are calculated as: $$ID=\log_{2}\left(\frac{2A}{W}\right)\;\mathrm{bits}$$
where *ID*—index of difficulty, *A*—amplitude, and *W*—widths. Each FTT trial was recorded as the time to complete in milliseconds (ms). An example of the FTT is shown in Figure 1. Before each FTT level, the subjects were introduced to the level and allowed to complete few practice runs. Based on the subject and investigator agreement of readiness, the official FTT trial was performed and recorded. The subjects were allowed a maximum of three mistakes during each level of difficulty and after the third mistake, the test was concluded and restarted. The study recorded a measure of the total average time of all 12 difficulty trials (hereafter referred as average total FTT time) and individual completion time for each of the FTT difficulties.

The hand dexterity and hand strength of the study subjects was determined using the O’Connor Dexterity test and a hand dynamometer, respectively [18]. The O’Connor dexterity test is a validated measure of the rapid manipulation of small objects. It requires participants to place one and three needle-like pins in each of the 100 holes (by picking up three pins at a time). Both one-pin and three-pin trials were performed once and recorded. After a practice of 30 holes and time to rest, the time of completion was computed as follows: $$Raw\;score=\frac{Time\;required\;for\;first\;half+\left(1.1\;\times \;time\;required\;for\;second\;half\right)\;}{2}$$

The raw time was recorded in seconds, where a longer time is indicative of worse performance. The hand grip strength was determined using a dynamometer and expressed on a scale between 0 and 100 kg. The final hand-grip strength was determined as an average from three consecutive trials and represented in kiloponds (kP).

Lastly, a Visual Reaction Time Test (VRTT) was performed using a Lafayette device that has eight specific light bulbs and eight corresponding reaction buttons. The visual stimuli are ordered in a semicircle around one additional central button. The participants are required to press and hold the central button until a random visual stimulus (1 out of 8 lights) is presented at random intervals. Once presented, the participant must release the central button and press the corresponding reaction button. Two automated chronometers were integrated and both started once the examiner imitated the visual stimuli. The first chronometer determined the time needed for the participant to recognize the stimuli and release the central button and the second chronometer determined the time needed to press the corresponding reaction button. Experimental situations included measurements of reaction times for one, two, and four light stimuli. To control the effects of mental and physical fatigue as extraneous variables, the experimental situations were rotated according to the Latin square principle. Each experimental situation was performed ten times within the Latin square principle and averaged for each subject. The average reaction time was recorded in milliseconds, where longer times indicate worse performance.

### 2.3. Patient-Reported Outcomes (PROs)

The general health status and health-related quality of life (QoL) was determined using the 36-Item Short Form Survey (SF-36) [19]. Permission for SF-36 use was acquired by the School of Public Health “Andrija Štampar” by University of Zagreb. The survey is composed of eight separate dimensions including: (1) physical functioning; (2) role limitations due to physical health; (3) role limitations due to emotional problems; (4) energy/fatigue; (5) emotional well-being; (6) social functioning; (7) bodily pain; and (8) general perception of health. All SF-36 dimensions are calculated from the subjective answers on a Likert-based question. Raw scores were transformed and can range between 0 and 100 (from worst to best outcome).

### 2.4. Statistical Analyses

All statistical analyses were performed using SPSS version 26.0 (IBM, Armonk, NY, USA). The data distribution and the distribution of the residuals was determined using visual inspection of the histograms and of the Q–Q plots. Additionally, the Kolmogorov–Smirnov test for data normality was performed. The comparison between the three study groups was performed using χ^2^ test (for categorical variables), Student’s *t*-test (for parametric numerical variables) and Mann–Whitney U-test (for nonparametric numerical variables). For the comparison of multiple groups, a one-way analysis of variance (ANOVA) and Kruskal–Wallis H-test were used appropriately. The intrarater and interrater reproducibility of the FTT was performed using Cronbach’s α coefficient and two-way mixed model interclass correlation coefficients (ICC). The pooled test–retest reproducibility over one week (including all segments of the FTT and the total average FTT performance) was plotted on Bland–Altman plots. The limits of agreement were determined based on data within the 5th–95th percentile. Based on the limits of agreement with percentage-based data, the data within the 5th and 95th percentiles were considered as the limits of a minimally detectable change (MDC). 

The ability of psychomotor performance to predict whether the subject was pwMS or HCs was determined using step-wise logistic regression model. In the models, the disease status was used as the dependent variable, whereas age, sex, and all psychomotor measures were used as independent predictors. The stepping criteria for variable entry was 0.05 and for variable removal was 0.1. To determine the best predictive variables, additional hypothesis-driven hierarchical regression models were used. The associations between the psychomotor performance of all three study groups with their patient-reported outcomes were determined using Pearson’s correlations. Additional heatmaps depicting the correlation coefficients were produced using GraphPad Prism version 8 (San Diego, CA, USA). The longitudinal analysis regarding psychomotor performance in 8 pwMS was assessed using a nonparametric paired Wilcoxon test. Line plots regarding the change in average FTT score and change in average dynamometer performance were produced. The effect size in nonparametric comparisons was calculated using: $$r=\frac{Z}{\sqrt{n}}$$
where *r* is the effect size, *Z* is the z-score, and *n* is the sample size. *p*-values lower than 0.05 were considered statistically significant.

## 3. Results

### 3.1. Demographic and Clinical Characteristics of the Study Groups

The demographic and clinical characteristics of the study groups is shown in Table 1. Based on the 1:1:1 matching criterion, there were no statistical differences between the pwMS, HCs, and pwMig in terms of sex, age, and formal years of education (*p* > 0.05). Moreover, all patients recruited into the study were right-hand dominant. Based on the SF-36 and when compared to HCs, the pwMS reported significantly greater impairments in physical functioning (median 70 vs. 97.5, Mann–Whitney U-test *p* = 0.005), greater role limitations due to physical health (median 75 vs. 100, Mann–Whitney U-test, *p* = 0.002) and lower general health (median 56 vs. 77, Mann–Whitney U-test, *p* < 0.001). Moreover, a composite physical health score was significantly lower in pwMS when compared to HCs (median 67.8 vs. 85.9, Mann–Whitney U-test, *p* < 0.001). 

### 3.2. Reproducibility of Fitts’ Tapping Task

The intrarater and interrater reproducibility of the FTT was performed twice in 5 HCs by two independent investigators. The overall reproducibility of the task was almost perfect with ICC of 0.961 (95% CI 0.88–0.995, *p* < 0.001) and Cronbach’s α of 0.961. The specific average times for each of the FTT difficulties acquired in the HCs are reported in Table 2. 

Moreover, the longitudinal reproducibility of the FTT was tested in 5 pwMS over a period of 1 week (tests performed twice by two independent raters at both timepoints) and are shown in Table 3. The % change of the total FTT time was only 1.66% (absolute range −0.11–3.1%). The Bland–Altman plots of the FTT reproducibility are shown in Figure 2 (the first panel depicts the absolute differences over the short follow-up period, whereas the second panel depicts the % differences between the timepoints). The average % difference from all acquired measures (individual FTT difficulties and a total/averaged FTT time) between the two timepoints was 1.74% (SD of 6.45) with 95% confidence intervals at 14.39% as the upper bound and −10.91% as the lower bound. All measures above the 95th percentile should be considered as an MDC. Based on the summed findings, a worsening of greater than 15% should be considered as an objectively true change. In our analysis, only one measurement from the individual FTT difficulty segments was at 22.88%. All remaining % differences were within −10–15% change. Additional Bland–Altman plots depicting % changes in the total FTT times (left) and only of the individual FTT difficulty segments (right) are shown in Figure 3. The total FTT times were similar with a 95% CI between 15.42% and −11.91% as the upper and lower bounds, respectively. In terms of total FTT times, the variation was significantly lower with a 95% CI between 4.2% and −0.89% as the upper and lower bounds, respectively.

### 3.3. Psychomotor Performance of the Study Groups

The psychomotor performance of the study groups is shown in Table 4. The pwMS needed significantly more time to complete all Fitts’ difficulties when compared to the HCs and pwMig (*p* < 0.005). Total average completion time for the Fitts’ task in pwMS was significantly higher when compared to HCs (18.2 s vs. 15.6 s, Mann–Whitney U-test *p* < 0.001). Similar findings were seen on the O’Connor Dexterity test, where pwMS had significantly lower hand dexterity performance with the 1 peg and 3 peg setting when compared to HCs (median 88.5 s vs. 80.4 s and 82.4 s, *p* = 0.007; median 312.4 s vs. 263.7 s and 267.2 s, *p* < 0.001). Lastly, the pwMS had significantly slower average reaction and decision times on the VRTT when compared to the HCs and pwMig (median 303.9 ms for pwMS vs. 223.7 ms for HCs and 236.9 ms pwMig, *p* = 0.001; median 381.8 ms for pwMS vs. 347.9 ms for HCs and 373.6 ms for pwMig, *p* = 0.007). On the contrary, there were no statistical differences in the dynamometer performance between the three study groups (*p* > 0.05). The differences between the three study groups in their psychomotor performance are also visualized in Figure 4.

In the step-wise hierarchical regression model, the FTT performance on the 4th (*p* = 0.002) and 6th difficulty (*p* = 0.03) together with the average O’Connor Dexterity test (*p* = 0.023) were the best differentiators between MS and HCs and were able to successfully classify the groups with 80% accuracy. In total, the three predictors were able to explain up to 58.6% of the data variance (Negelkerke R^2^ = 0.586). Similar findings were seen in the step-wise regression model where the performance of the 4th FTT difficulty (*p* = 0.003) and O’Connor dexterity (*p* = 0.01) were best differentiators between pwMS and pwMig (Negelkerge R^2^ = 0.5).

The hypothesis-driven hierarchal regression model is shown in Table 5. In addition to age and sex as covariates, the model incorporates the measures with significant differences between pwMS and HCs from the direct comparison. Three consecutive additions of the average FTT time, average O’Connor dexterity time, and average decision time improved the predictivity of the model (Negelkerge R^2^ change from 0.284 > 0.454 > 0.512). The last addition of average decision time did not significantly improve the predictivity (Negelkerge R^2^ change from 0.512 to 0.534).

### 3.4. Relationship between Psychomotor Performance and Patient-Reported Outcomes

The correlations between the psychomotor performance and SF-36 in the pwMS and HCs are shown in Figure 5A and Figure 5B, respectively. In the pwMS, the higher average FTT time was associated with more role limitations due to physical health (r = −0.443, *p* = 0.014) and overall physical health (r = −0.382, *p* = 0.032). Moreover, the longer average decision time in VRTT was associated with poorer outcomes as measured by role limitations due to physical health (r = −0.385, *p* = 0.036), role limitations due to emotional problems (r = −0.435, *p* = 0.007), social functioning (r = −0.467, *p* = 0.009), and mental health (r = −0.492, *p* = 0.006). In the HCs, the FTT time was not associated with any PRO measures.

Lastly, the average FTT time was significantly associated with multiple PRO measures in the pwMig. In particular, a longer FTT time was associated with worse role limitations due to emotional problems (r = −0.365, *p* = 0.047), mental health (r = −0.501, *p* = 0.005), general health perception (r = −0.444, *p* = 0.014), and overall physical health (r = −0.476, *p* = 0.008). In contrast to the other groups, the average reaction time in pwMig was associated with greater role limitation due to emotional problems (r = −0.459, *p* = 0.011), social functioning (r = −0.416, *p* = 0.022), and overall physical (r = 0.426, *p* = 0.019) and mental health (r = −0.459, *p* = 0.011).

### 3.5. Longitudinal Changes in Psychomotor Performance in pwMS

Eight pwMS that were originally enrolled in this study had a clinical examination at 2-year follow-up. The EDSS of these 8 pwMS remained stable over the follow-up period (median baseline EDSS of 1.0 vs. 1.0, *p* = 1.000), and none of the pwMS had new/enlarging T2-FLAIR hyperintensities. Despite their clinical stability, the cohort of 8 pwMS experienced significant slowing in their average FTT scores (mean 14.5 s to 17.8 s, paired Wilcoxon test *p* = 0.012). Based on the 15% cut-off value for pathological changes in FTT scores, 6 out 8 (75.0%) of pwMS experienced a worsening in their psychomotor performance. In contrast, the pwMS did not experience any change in their motor strength as measured by the dynamometer performance (*p* > 0.05). A visual representation of the changes in average FTT and dynamometer performance in these 8 pwMS are shown in Figure 6. 

## 4. Discussion

The findings from this validation study are multifold. First, FTT is highly reproducible, with almost perfect intrarater and interrater reproducibility. Second, an FTT change of 15% may be considered as the cut-off for minimal clinically-detectable change and utilized in the assessment of worsening in psychomotor performance. Lastly, FTT can differentiate newly-diagnosed pwMS from HCs and majority of clinically-stable pwMS experience a worsening in psychomotor performance (as measured by FTT) over 2 years. The importance of these findings and their relevance in the care of pwMS are discussed hereafter. 

Multiple cut-offs regarding the 9HPT performance have been proposed for both diagnostic and prognostic purposes. They are expressed either as deviation from age and sex-adjusted normative values (either 1 SD or 1.95 SD), the number of pegs completed per second (0.5 pegs/s or 0.27 pegs/s) or in absolute time measures such as a cut-off of 18 s [4,20]. A worsening of 20% has been commonly utilized to represent a clinically-meaningful change in 9HPT [21]. However, studies suggest that 9HPT is not sensitive enough to detect subtle impairments in mildly-disabled pwMS [22]. To capture the aforementioned 20% change, the minimal baseline 9HPT performance should be greater than 21 s [22]. A substantial portion of the pwMS with mild disability (EDSS > 3.0) will perform below the cut-off and be influenced by a floor effect [22]. With respect to the FTT, we observed a similar 15% change that can be utilized for tracking the intraindividual worsening in psychomotor performance. To expand the utility of FTT, future studies should also aim at developing age- and sex-based normative values that would allow the creation of z-scores, the use of standard deviations, and the determination of the floor/celling effects of various subpopulations. Moreover, studies should determine to what extent the performance on the FTT is driven by pure manual dexterity and by cognitive slowing. Unlike in the HCs, the FTT performance in our MS sample was not significantly associated with the O’Connor dexterity test, with both FTT and hand dexterity having an independent diagnostic value. While these correlation discrepancies could be argued to be suggestive of two separate processes affecting the FTT performance in pwMS, another properly-designed study should test this hypothesis. Lastly, the lack of differences in dynamometer outcomes between pwMS and HCs and lack of a correlation between dynamometer and FTT strongly suggests that the FTT performance in newly-diagnosed MS is not affected by motor/hand weakness, and is more likely related to a centralized representation of psychomotor processing per se. As with other cognitive and hand dexterity tests, visual acuity is one confounder that should be additionally considered [23].

The subclinical worsening of hand dexterity and psychomotor/cognitive performance in newly-diagnosed pwMS has been demonstrated in the literature [7]. From a convenience sample of 1091 pwMS with a low median EDSS score of 2.0, 37.4% experienced a clinically-detectable worsening and a smaller portion of pwMS (4.0%) experienced isolated cognitive worsening over 12 months [24]. An increasing number of studies demonstrate subtle and subclinical worsening of multiple neurological functions in people that will be diagnosed with MS in the future (prodromal MS) [25]. For example, first-degree family members of pwMS that are at high risk for developing the disease themselves demonstrate subclinical worsening in vibration sensitivity [26]. Similar neurological changes are seen in RIS where up to 1/3 of cases demonstrate cognitive impairment or a worsening of hand function (tapping) [7,27].

The significant associations between FTT performance and PROs in the pwMS provide some indication of the construct and ecological validity of the test. Despite having minimal MS disability in our study sample, the FTT measure was associated with patient-reported limitations due to physical health. These results corroborate other literature findings, where hand dexterity (measured by 9HPT) has been associated with PRO measures of perceived hand impairment such as ABILHAND, manual ability measure (MAM-36), and motor activity log (MAL) [28,29,30]. The associations between VRTT performance and a greater range of physical and mental limitations confirm the higher cognitive load of this particular test. Interestingly, and contrarily to the pwMS, the FTT performance in the pwMig was associated with greater perception of mental limitations. Such a relationship is well known in the literature, corroborating the link between migraine and its effect on mental health factors such as depression, stress, and anxiety [31,32].

Our study has several limitations that we ought to outline. First, the longitudinal aspect of the study involved a very limited number of pwMS and no HCs. Future FTT studies should determine the expected rate of decline in performance in HCs as a function of aging. Moreover, the study did not include 9HPT and SDMT as a direct comparator and current gold standard for measuring hand dexterity and cognitive processing speed. A direct comparison to 9HPT would allow the derivation of final criterion validity. There is also a need to determine the long-term FTT performance in pwMS with greater physical disability which may influence the extent of test–retest variability. Studies should also determine to what extent the FTT performance is influenced by cognitive and motor processes and whether FTT can be an early indicator for cognitive changes in MS. While migraine was not the direct focus of this investigation, future studies should utilize these preliminary findings and determine the longitudinal utility of FTT in pwMig and its responsiveness to disease changes/treatment.

## 5. Conclusions

FTT is a highly-reproducible test for measuring psychomotor performance in newly-diagnosed pwMS. Based on test–retest metrics, a cut-off of 15% worsening in FTT should be considered as a minimally-detectable change. Despite being clinically stable, a majority of newly-diagnosed pwMS experience a worsening in FTT performance. The performance on FTT may be a good indicator of impairments affecting functions that incorporate multiple aspects such as motor and cognitive processing, hand dexterity, and reaction time. 

## Figures and Tables

**Figure 1 medicina-59-00029-f001:**
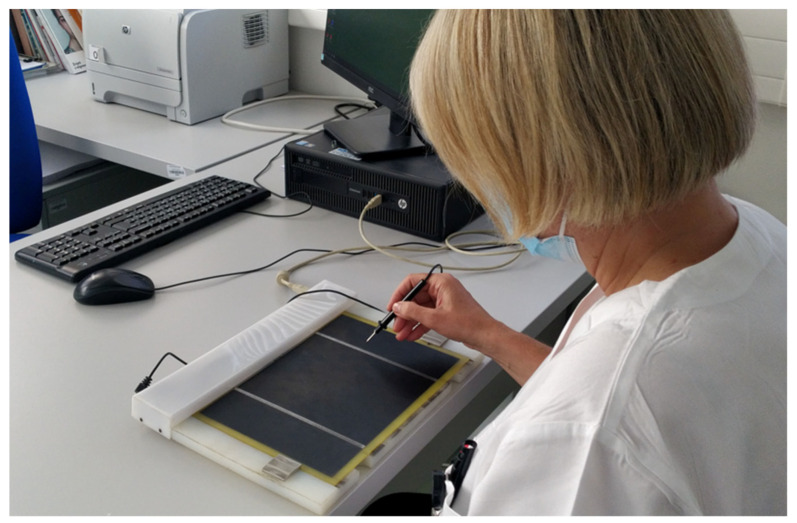
Example of Fitts’ Tapping Task applied in a real-world setting. Legend: The subject is instructed to alternate and touch the strip-targets using the stylus. Trials included different difficulties based on the target widths of 0.5 cm, 1.0 cm, 2.0 cm, and 4.0 cm, and movement amplitudes of 4.0 cm, 8.0 cm, 16.0 cm, respectively.

**Figure 2 medicina-59-00029-f002:**
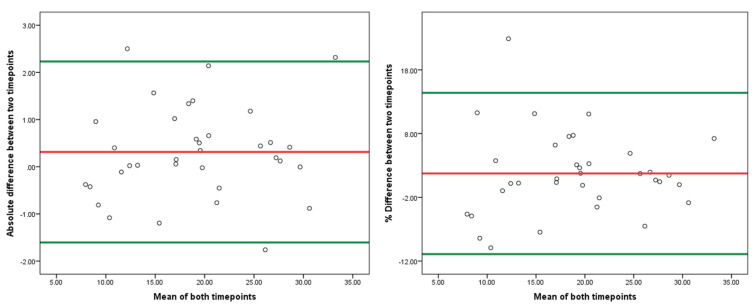
Bland–Altman plots depicting the reproducibility of the FTT including all measures (6 Fitts’ difficulties) and the average FTT score (average of the 12 trials together). Legend: FTT—Fitts’ Tapping Task. Left—absolute difference in Fitts’ Tapping Tasks within the people with multiple sclerosis over 1-week follow-up, Right—percent difference in Fitts’ Tapping Tasks within the people with multiple sclerosis over 1-week of follow-up. Each dot represents the difference between the two timepoints over 1-week. The green lines represent the lower and upper 95% confidence intervals and the red line represents the mean value.

**Figure 3 medicina-59-00029-f003:**
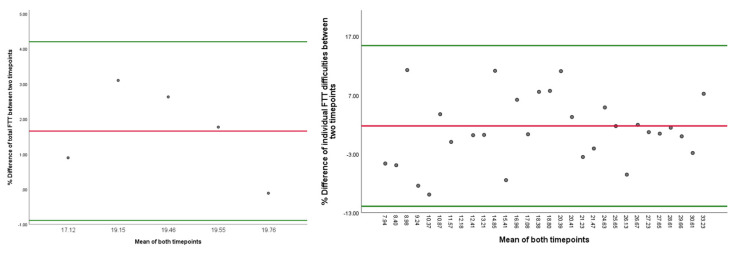
Bland–Altman plots depicting the reproducibility of total FTT time and individual FTT difficulties. Legend: FTT—Fitts’ Tapping Task. Left—absolute difference in Fitts’ Tapping Tasks within the people with multiple sclerosis over 1-week follow-up, Right—percent difference in Fitts’ Tapping Tasks within the people with multiple sclerosis over 1-week of follow-up. Each dot represents the difference between the two timepoints over 1-week. The green lines represent the lower and upper 95% confidence intervals and the red line represents the mean value.

**Figure 4 medicina-59-00029-f004:**
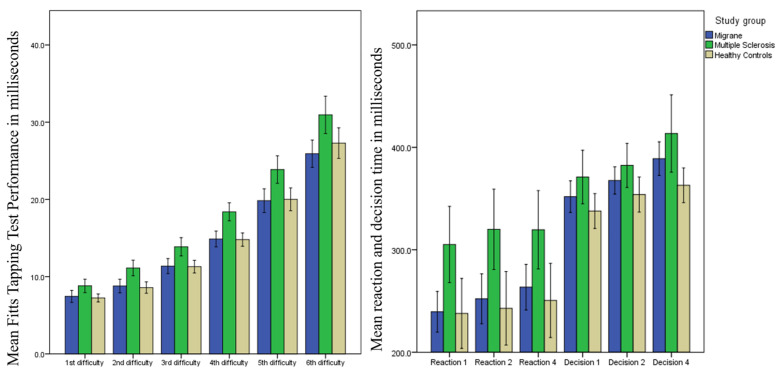
Box plot illustrating psychomotor performance between the three study groups. Legend: FTT—Fitts’ Tapping Task. The FTT performance measured at each difficulty level is shown in the first panel (left), whereas the reaction and decision times are shown in the second panel (right). Data are shown as the mean and error bars depict two standard errors.

**Figure 5 medicina-59-00029-f005:**
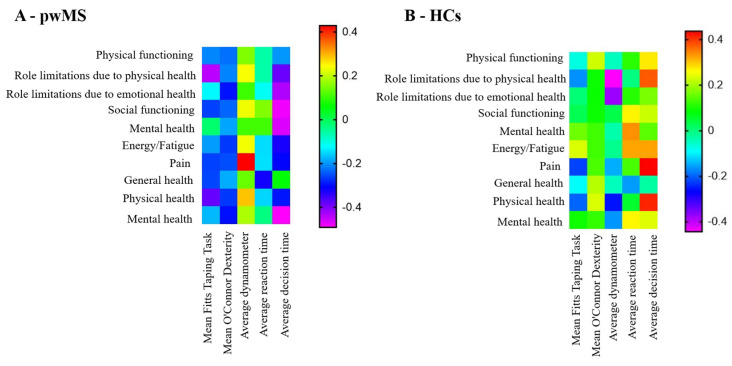
Correlation matrix between psychomotor performance and PROs in pwMS and HCs. Legend: pwMS—people with multiple sclerosis, HCs—healthy controls, PROs—patient-reported outcomes. The correlation matrix depicts the correlation coefficients (r) derived from Pearson’s correlations. (**A**)—correlations between psychomotor performance and PROs in people with multiple sclerosis, (**B**)—correlations between psychomotor performance and PROs in healthy controls.

**Figure 6 medicina-59-00029-f006:**
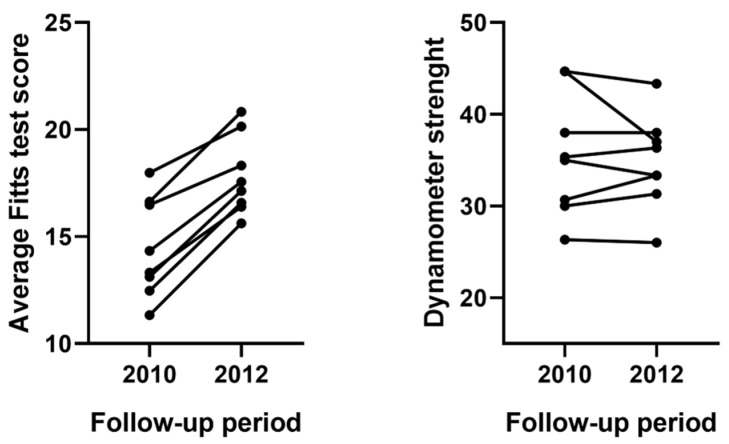
Change in psychomotor (FTT) and physical hand (dynamometer) performance over 2-year follow-up in eight pwMS. Legend: FTT—Fitts’ Tapping Task, pwMS—people with multiple sclerosis. Paired *t*-test was used to calculate the intraindividual change in Fitts’ Tapping Task and dynamometer measures. Each line represents one pwMS that was followed over 2-years.

**Table 1 medicina-59-00029-t001:** Demographic and patient-based outcomes in the study groups.

Demographic and PRO Measures	PwMS (*n* = 30)	Migraine (*n* = 30)	HCs (*n* = 30)	Across Groups *p*-Value	MS vs. HCs *p*-Value
Female, *n* (%)	22 (73.3)	22 (73.3)	22 (73.3)	1.000	1.000
Age, mean (SD)	37.3 (9.7)	39.9 (7.5)	37.5 (6.8)	0.389	1.000
Formal years of education, *n* (%)					
High school completed	21 (70.0)	15 (50.0)	14 (46.7)	0.333	0.185
Bachelor’s degree completed	6 (20.0)	12 (40.0)	11 (36.7)		
Postgraduate degree completed	3 (10.0)	3 (10.0)	5 (16.7)		
36-Item Short Form Survey (SF-36), median (IQR)					
Physical functioning	70 (43.8–96.3)	90 (65.0–100.0)	97.5 (83.8–100.0)	**0.018**	**0.005**
Role limitations due to physical health	75 (25–100)	75 (25.0–100.0)	100 (100.0–100.0)	**0.001**	**0.002**
Role limitations due to emotional problems	83.3 (33.3–100)	100 (33.3–100.0)	100 (66.7–100.0)	0.259	0.087
Energy/fatigue	57.5 (40.0–70.0)	55 (35.0–66.3)	52.5 (45.0–75.0)	0.72	0.829
Emotional well-being	70 (59.0–80.0)	68 (55.0–77.0)	68 (59.0–81.0)	0.719	0.917
Social functioning	75.0 (62.5–100.0)	75 (37.5–87.5)	81.3 (59.4–100.0)	0.294	0.934
Pain	70 (38.8–72.0)	60 (37.5–72.5)	80 (60.0–90.0)	**0.019**	0.271
General health perception	56 (38.8–72.0)	63.5 (45.8–77.0)	77 (69.3–87.8)	**<0.001**	**<0.001**
Physical health	67.8 (41.6–83.3)	68 (41.4–82.1)	85.9 (74.3–91.8)	**<0.001**	**0.001**
Mental health	69.9 (49.5–82.8)	72.1 (48.2–82.1)	74.4 (62.2–86.8)	0.486	0.506

Legend: PRO—patient-reported outcomes, PwMS—patients with multiple sclerosis, HCs—healthy controls, SD—standard deviation, SF-36—36-Item Short Form Survey, IQR—interquartile range. The three groups were compared using a one-way analysis of variance (ANOVA) for normally-distributed variables and the Kruskal–Wallis H test for nonparametric data. The comparisons between pwMS and HCs was performed using Student’s *t*-test and Mann–Whitney U test as appropriate. *p*-values lower than 0.05 were considered statistically significant and are shown in bold.

**Table 2 medicina-59-00029-t002:** Intrarater and interrater reproducibility of the FTT in healthy controls.

FTT Performance	Test	Rater	Mean (SD)
1st difficulty	Test 1	Rater 1	7.53 (0.99)
		Rater 2	7.48 (0.99)
	Test 2	Rater 1	7.52 (0.34)
		Rater 2	7.41 (0.36)
2nd difficulty	Test 1	Rater 1	9.7 (0.99)
		Rater 2	9.73 (0.93)
	Test 2	Rater 1	9.66 (1.12)
		Rater 2	9.62 (1.04)
3rd difficulty	Test 1	Rater 1	12.72 (1.83)
		Rater 2	12.81 (1.75)
	Test 2	Rater 1	13.77 (3.12)
		Rater 2	13.59 (2.99)
4th difficulty	Test 1	Rater 1	17.73 (1.52)
		Rater 2	17.71 (1.53)
	Test 2	Rater 1	18.18 (2.58)
		Rater 2	18.21 (2.58)
5th difficulty	Test 1	Rater 1	26.41 (3.69)
		Rater 2	26.43 (3.55)
	Test 2	Rater 1	25.42 (3.25)
		Rater 2	25.48 (3.18)
6th difficulty	Test 1	Rater 1	30.11 (3.37)
		Rater 2	29.78 (3.13)
	Test 2	Rater 1	30.98 (3.32)
		Rater 2	30.93 (3.23)

Legend: FTT—Fitts’ Tapping Task, SD—standard deviation.

**Table 3 medicina-59-00029-t003:** Short-term reproducibility of Fitts’ Tapping Task in five pwMS over one week.

Longitudinal FTT Change (*n* = 5)	Median Total Time (msec)	ICC *p*-Value ^a^	Mean (Median) Raw Change (msec)	Mean (Median) % Change	Min–Max Range of Change (%)
Average of all 4 trials at Visit 1	19.2 (17.9–19.6)	**0.001**	0.31 (0.34)	1.66 (1.77)	−0.11–3.1
Average of all 4 trials at Visit 2	19.7 (18.3–19.7)				

Legend: FTT—Fitts’ Tapping Task. Average FTT times are reported as median (interquartile range). PwMS—people with multiple sclerosis, ICC—interclass correlation coefficient, msec—milliseconds, Min—minimum, Max—maximum. ^a^ Trials compared using Wilcoxon Signed Ranks Test. *p*-values lower than 0.05 were considered statistically significant and shown in bold.

**Table 4 medicina-59-00029-t004:** Psychomotor performance in the study groups.

Psychomotor Tests	PwMS (*n* = 30)	PwMig (*n* = 30)	HCs (*n* = 30)	Across Groups *p*-Value	Effect Size for pwMS vs. HCs	PwMS vs. HCs *p*-Value
**Average Fitts’ Tapping Task time (seconds)**	18.2 (16.0–19.7)	14.8 (12.4–17.2)	15.6 (12.3–17.0)	**<0.001**	0.476	**<0.001**
1st difficulty time	8.9 (7.2–9.8)	7.1 (5.7–9)	7.2 (6.1–8.3)	**0.005**	0.387	**0.003**
2nd difficulty time	10.8 (9.4–12.5)	8.3 (6.7–11.1)	8.6 (6.8–10.0)	**0.001**	0.456	**<0.001**
3rd difficulty time	13.8 (12.1–15.8)	11.1 (9.6–13.8)	11.4 (9.5–13.0)	**0.002**	0.411	**0.001**
4th difficulty time	18.2 (16.9–19.7)	14.3 (12.375–17)	15.2 (12.8–16.6)	**<0.001**	0.58	**<0.001**
5th difficulty time	25.1 (20.1–27.7)	19.4 (16.1–23.6)	20.6 (15.9–23.2)	**0.001**	0.389	**0.003**
6th difficulty time	33.3 (25–35.2)	27.7 (21.9–30)	28.6 (22.3–31.8)	**0.002**	0.326	**0.011**
**Average O’Connor Dexterity (seconds)**	201.4 (177.9–248.0)	174.5 (165.2–197.9)	173.7 (161.9–194.0)	**<0.001**	0.499	**<0.001**
1 peg	88.5 (82.5–98.9)	82.4 (76.3–92.9)	80.4 (74.9–86.5)	**0.007**	0.407	**0.002**
3 pegs	312.4 (270.1–378.1)	267.2 (255.6–307.9)	263.7 (248.9–300.7)	**<0.001**	0.457	**<0.001**
**Average dynamometer (kP)**	33.3 (28.3–44.4)	37.7 (30.2–46.7)	37.3 (32.3–46.9)	0.247	0.217	0.093
1st attempt	33 (29.0–43.5)	38.5 (31.5–49.5)	39 (31.8–47)	0.233	0.207	0.108
2nd attempt	33 (28.0–44.3)	38 (29.8–45.5)	36.5 (32.0–48.0)	0.284	0.214	0.097
3rd attempt	32 (26.8–45.0)	37.5 (30.0–45.8)	36 (33.8–46.5)	0.292	0.191	0.139
**Average reaction time (ms)**	303.9 (243.9–361.6)	236.9 (218.3–286.1)	223.7 (198.8–254.2)	**0.001**	0.468	**<0.001**
1 stimulus	291.3 (230.6–361.2)	226.2 (202.6–271.7)	218.4 (200.2–252.2)	**0.001**	0.437	**0.001**
2 stimuli	311.8 (248.9–386.1)	233.5 (211.5–283.1)	225.1 (194.7–260.7)	**0.001**	0.443	**0.001**
4 stimuli	310.3 (254.6–367.3)	245.8 (227.3–312.5)	229.1 (195.8–274.6)	**0.003**	0.414	**0.001**
**Average decision time (ms)**	381.8 (359.8–413.5)	373.6 (339.4–392.1)	347.9 (314.4–363.7)	**0.007**	0.389	**0.003**
1 stimulus	365.25 (338.2–389.1)	338.4 (320.1–381.1)	327.7 (304.5–354.7)	**0.02**	0.355	**0.006**
2 stimuli	385.25 (336.8–407.4)	363.9 (343.9–384.5)	350.4 (313.9–374.9)	**0.03**	0.323	**0.012**
4 stimuli	402.75 (367.1–442.1)	394.4 (355.5–420.9)	352.4 (331.1–382.0)	**0.008**	0.373	**0.004**

Legend: PwMS—patients with multiple sclerosis, PwMig—people with migraine, HCs—healthy controls, kP—kilopond, ms—milliseconds. The Kruskal–Wallis H-test was used for comparison of all three groups and Mann–Whitney U test was used for comparison between PwMS and HCs. *p*-values lower than 0.05 were considered statistically significant and are shown in bold. All data are shown as the median (interquartile range).

**Table 5 medicina-59-00029-t005:** Hypothesis-driven hierarchical binary regression model for differentiation between pwMS and HCs.

	Predictor Added	B	S.E.	Wald	*p*-Value	Exp(B)
Step 1	Age	0.081	0.053	2.334	0.127	1.084
	Sex	−0.13	0.842	0.024	0.877	0.878
Step 2	Average FTT time	−0.27	0.138	3.846	0.049	0.763
Step 3	Average O’Connor time	−0.037	0.016	5.538	0.019	0.964
Step 4	Average reaction time	−0.007	0.004	3.269	0.071	0.993
Step 5	Average decision time	−0.009	0.007	1.552	0.213	0.991

Legend: pwMS—people with multiple sclerosis, HCs—healthy controls, B—beta, S.E.—standard error, FTT—Fitts’ Tapping Task. *p*-values lower than 0.05 were consider statistically significant and are shown in bold.

## Data Availability

The data presented in this study are available on request from the corresponding author.

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
