# Peer review of "Fitts’ Tapping Task as a New Test for Cognition and Manual Dexterity in Multiple Sclerosis: Validation Study"

_medicina, 2022, doi:10.3390/medicina59010029_

Round 1

Reviewer 1 Report

This is an interesting exploration of a novel instrument for use in pwMS. I think this is a relevant topic, and your manuscript touches on a few different aspects of the instrument (reproducibility, use as a classifier, FTT correlations in pwMS vs in HC). I have some clarifying questions, mostly to help me make sense of what is being reported from the FTT (between 12 tasks, 6 difficulty levels, an overall/summary score, etc.).  

Is it called the Fitts Tapping Test or the Fitts Tapping Task? Be consistent.

Criteria for newly-diagnosed MS is different in the abstract (within 2 years) than what is given in the paper (within 5 years).

Line 134: 4.0 cm is listed twice for target widths (and also in the caption of fig 1). Is this intentional?

It’s hard to follow how the different parts/tasks in the FTT relate to each other. There are a number of terms used, where I can’t tell if they refer to the same or different aspects of the FTT. There are 12 tasks, which relate to 6 difficulty levels (where 2A/W can equal 1/16, 1/8, ¼, ½, 1, or 2). Is a “trial” the time taken to complete all 12 tasks, or is each task a “trial”? Are tasks of the same difficulty averaged together for reporting?

Why was the longitudinal analysis conducted on only 8 pwMS? Your methods section only describes the 3 groups of 30 people each. Please describe how and why other analytical samples were chosen (these 8, as well as the 5 HC and 5 pwMS selected for the reproducibility studies).

Lines 224-228: make it clear what groups you are comparing with these results, since you have three groups. The table indicates that this is pwMS vs HC, but you should make this clear in the text, too.

Section 3.2: I don’t see a Supplement Table 1. What was assessed for reproducibility in the HC? Each of the 12 tasks, each of the 6 difficulties, an overall/average score, some combination of these?

It’s not quite clear to me what measures are being reported for the FTT results. My understanding is that the FTT is made up of 12 tasks (4 widths x 3 amplitudes). I also see reference to average Fitts score (in caption for figure 2) and total average completion time. Can you clarify what these are? Is the total score time taken to complete all tasks, or an averaging of scores on the 12 tasks? It looks like table 2 is probably reporting a summary score (rather than all task measurements), since the max change is reported as 3.1%, but you state in the text that there was a difference of 22.88% for one of the “difficulty segments”. Similarly, the average % difference is given in the text as 1.74, but in the table as mean=1.66/median=1.77. Are these referring to different measures – i.e. one referring to the average of all tasks, the other referring to the average of a summary/total score? Please clarify what is being presented in your results.

As you’ve defined MDC as a worsening <15%: does this refer to a worsening of <15% in any of the measures/tasks/difficulty levels, or only in the total/average/summary FTT score?

Per your caption to figure 2, it looks like all measures, as well as average Fitts score, are on the Bland Altman plots – meaning that each person is represented on the plot multiple times. It looks like this must be the 6 difficulty levels (not the 12 tasks), correct? Please make more clear what is being presented. Also – if I’m interpreting this correctly – I don’t know if this is a valid way to display results. Including each person more than once on the plot violates the independence assumption of the Bland Altman plots. There is correlation between each person’s performance on the individual measures within the FTT. Also, I’ve never seen multiple measures/submeasures included on the same BA plot before (the exception being for two measures, where one measure is x-axis and one measure is y-axis). I think a more valid approach would be to present separate BA plots for the tasks you want to report on. A change of 22% will have a different meaning if it is on the total score vs one of the specific tasks/levels and this should be made clear. If you are wedded to including all tasks/levels on one plot, and using all of them to define your MDC, please make this clear in the text and justify your decision. Distinguishing individual people on the plots by color-coding, or symbol shape, would also be helpful and more honest. 

Line 268: completion time was significantly higher (not lower) in pwMS, correct?

When you report in the text the scores for all three groups (pwMS, pwMig, and HC), make sure you keep them in the same order for consistency. It looks like you reported pwMS, pwMig, HC in lines 270-1, pwMS, HC, pwMig in lines 273-4.

Clarify here the difference between your two stepwise models (lines 288-295). The first one mentioned, that selected the 4th and 6th difficulty and the average O’Connor Dexterity test – it looks like this was to distinguish pwMS from HC only, correct? (rather than distinguishing from HC and pwMig combined, i.e. the full sample). Please make it clear in the text what is being compared with each model.  

It doesn’t look like figure 4C is shown; remove mention in text. Also, text indicates that 4A is HC, but caption indicates that 4A is pwMS.

Some typos:

Fitts law --> Fitts' law (line 74)

studies --> studied (line 86)

attach --> attack (line 118)

stepping --> tapping (line 242)

logistical --> logistic (line 288)

hierarchal --> hierarchical (line 296)

bellow --> below (line 366)

Author Response

Reviewer 1:

This is an interesting exploration of a novel instrument for use in pwMS. I think this is a relevant topic, and your manuscript touches on a few different aspects of the instrument (reproducibility, use as a classifier, FTT correlations in pwMS vs in HC). I have some clarifying questions, mostly to help me make sense of what is being reported from the FTT (between 12 tasks, 6 difficulty levels, an overall/summary score, etc.).  

Response: We thank the Reviewer for providing excellent comments and suggestions that have significantly improved the quality and readability of our manuscript. Point-by-point answers to all comments are shown hereafter.

Is it called the Fitts Tapping Test or the Fitts Tapping Task? Be consistent.

Response: This has been standardized through the manuscript. We corrected it to Fitts Tapping Task as mentioned in previous literature.

Criteria for newly-diagnosed MS is different in the abstract (within 2 years) than what is given in the paper (within 5 years).

Response: We thank the Reviewer for noticing this discrepancy. This has been corrected as 2-years within diagnosis throughout the manuscript.

Line 134: 4.0 cm is listed twice for target widths (and also in the caption of fig 1). Is this intentional?

Response: One of the “4.0cm” repeat was corrected.

It’s hard to follow how the different parts/tasks in the FTT relate to each other. There are a number of terms used, where I can’t tell if they refer to the same or different aspects of the FTT. There are 12 tasks, which relate to 6 difficulty levels (where 2A/W can equal 1/16, 1/8, ¼, ½, 1, or 2). Is a “trial” the time taken to complete all 12 tasks, or is each task a “trial”? Are tasks of the same difficulty averaged together for reporting?

Response: These are all excellent points. We have further clarified the tasks and corresponding times that are reported in the manuscript.

Why was the longitudinal analysis conducted on only 8 pwMS? Your methods section only describes the 3 groups of 30 people each. Please describe how and why other analytical samples were chosen (these 8, as well as the 5 HC and 5 pwMS selected for the reproducibility studies).

Response: This is an excellent suggestion. We have clarified that there were essentially three different cohorts that are mentioned in the manuscript. This is now at line 134-137.

  • Only 8 out of the 30 pwMS were available and agreed to return for the 2-year follow-up examination. This has been clarified in the manuscript.
  • Additional 5 HCs were recruited for the purpose of intra-rater and inter-rater reproducibility without the effect of a disease interfering.
  • Additional and different 5 pwMS were acquired only for test-retest reproducibility acquired at index and after 7 days.

Lines 224-228: make it clear what groups you are comparing with these results, since you have three groups. The table indicates that this is pwMS vs HC, but you should make this clear in the text, too.

Response: We have specified the groups for the comparisons as suggested.

Section 3.2: I don’t see a Supplement Table 1. What was assessed for reproducibility in the HC? Each of the 12 tasks, each of the 6 difficulties, an overall/average score, some combination of these?

Response: For greater comprehensiveness and clarity, we decided to move the supplement material into the main text. This would allow readers to have direct access to all material at one place. This is now added as Table 2 and all remaining Tables are re-numbered accordingly.

It’s not quite clear to me what measures are being reported for the FTT results. My understanding is that the FTT is made up of 12 tasks (4 widths x 3 amplitudes). I also see reference to average Fitts score (in caption for figure 2) and total average completion time. Can you clarify what these are? Is the total score time taken to complete all tasks, or an averaging of scores on the 12 tasks? It looks like table 2 is probably reporting a summary score (rather than all task measurements), since the max change is reported as 3.1%, but you state in the text that there was a difference of 22.88% for one of the “difficulty segments”. Similarly, the average % difference is given in the text as 1.74, but in the table as mean=1.66/median=1.77. Are these referring to different measures – i.e. one referring to the average of all tasks, the other referring to the average of a summary/total score? Please clarify what is being presented in your results.

Response: The average FTT time is an average of all difficulty tasks (12 trials, 6 difficulties). This has been demonstrated in the Table with the psychomotor performance of the groups. The average FTT of 18.2s was derived by adding all 6 difficulties together and then dividing the time with 6. This has been further explained in the Methods of the manuscript. The Table with the short-term reproducibility presents the average of the 6 difficulties (12 trials).

As you’ve defined MDC as a worsening <15%: does this refer to a worsening of <15% in any of the measures/tasks/difficulty levels, or only in the total/average/summary FTT score?

Response: This is an excellent point. We have further clarified the data in the text and specified which differences are being referred. 

Per your caption to figure 2, it looks like all measures, as well as average Fitts score, are on the Bland Altman plots – meaning that each person is represented on the plot multiple times. It looks like this must be the 6 difficulty levels (not the 12 tasks), correct? Please make more clear what is being presented. Also – if I’m interpreting this correctly – I don’t know if this is a valid way to display results. Including each person more than once on the plot violates the independence assumption of the Bland Altman plots. There is correlation between each person’s performance on the individual measures within the FTT. Also, I’ve never seen multiple measures/submeasures included on the same BA plot before (the exception being for two measures, where one measure is x-axis and one measure is y-axis). I think a more valid approach would be to present separate BA plots for the tasks you want to report on. A change of 22% will have a different meaning if it is on the total score vs one of the specific tasks/levels and this should be made clear. If you are wedded to including all tasks/levels on one plot, and using all of them to define your MDC, please make this clear in the text and justify your decision. Distinguishing individual people on the plots by color-coding, or symbol shape, would also be helpful and more honest. 

Response: We agree with the Reviewer. The existing plot uses the 6 difficulty times and an average of them as total FTT. (In total there are 7 points in the chart that are derived from one pwMS with total of 35 contributing points). In order to ensure total transparency, we have now separated the plots for the difficulties only (1 plot with the 6 difficulties) and a new plot with only the average FTT time (only 5 data points for the 5 pwMS).

Line 268: completion time was significantly higher (not lower) in pwMS, correct?

Response: We thank the Reviewer for pin-pointing this and other mistakes. We have corrected this accordingly.

When you report in the text the scores for all three groups (pwMS, pwMig, and HC), make sure you keep them in the same order for consistency. It looks like you reported pwMS, pwMig, HC in lines 270-1, pwMS, HC, pwMig in lines 273-4.

Response: The ordering of the comparisons has been corrected as suggested. This has significantly improved the manuscript readabiltiy.

Clarify here the difference between your two stepwise models (lines 288-295). The first one mentioned, that selected the 4th and 6th difficulty and the average O’Connor Dexterity test – it looks like this was to distinguish pwMS from HC only, correct? (rather than distinguishing from HC and pwMig combined, i.e. the full sample). Please make it clear in the text what is being compared with each model.  

Response: This is correct. The first model compared MS vs. HCs where the second model compared MS vs. pwMig. This has been specified in the Results section.

It doesn’t look like figure 4C is shown; remove mention in text. Also, text indicates that 4A is HC, but caption indicates that 4A is pwMS.

Response: These mistakes have been corrected accordingly.

Some typos:

Fitts law --> Fitts' law (line 74)

studies --> studied (line 86)

attach --> attack (line 118)

stepping --> tapping (line 242)

logistical --> logistic (line 288)

hierarchal --> hierarchical (line 296)

bellow --> below (line 366)

Response: We thank the Reviewer for the careful read-through of the manuscript. This has been all corrected accordingly.

Reviewer 2 Report

In this paper, Glavor and colleagues studied thirty newly-diagnosed pwMS with low disability,, 30 pwMig and 30 healthy controls. These subjects underwent a psychomotor assessment using the Fitts Tapping Task 23 (FTT), O’Connor hand dexterity test, and Visual Reaction Time Test (VRTT). They also tested hand strenght with a hand-grip dynamometer and collected the patient-reported outcomes. Eight pwMS returned for the same test procedures 2-years after baseline. 

FTT was demonstrated to have a high intrarater and interrater reproducibility with a test-retest MDC of 15%. PwMS had significantly slower FTT time and O’Connor dexterity time when compared to pwMig and HCs and higher Fitts difficulty levels (4th and 6th difficulty) and average performance on O’Connor test were able to differentiate newly-diagnosed pwMS from HCs with 80% accuracy (p<0.01). Slower FTT performance was correlated with worse PROs due to physical health. After a 2-year follow-up, despite being clinically stable, 6 out of 8 pwMS had more than 15% worsening in their average FTT time. Therefore the Authors conclude that FTT represents a highly reproducible test for measuring psychomotor performance in newly-diagnosed pwMS and can capture insidious worsening in psychomotor performance and cognitive function in early stages of the disease.

In my opinion, this is a well executed and novel study and adds to our knowledge about the tests available for measuring psychomotor performance in the early stages of MS. The limitations are recognized and discussed.

There are few English and syntax errors throughout that could be corrected.

Author Response

In this paper, Glavor and colleagues studied thirty newly-diagnosed pwMS with low disability,, 30 pwMig and 30 healthy controls. These subjects underwent a psychomotor assessment using the Fitts Tapping Task 23 (FTT), O’Connor hand dexterity test, and Visual Reaction Time Test (VRTT). They also tested hand strenght with a hand-grip dynamometer and collected the patient-reported outcomes. Eight pwMS returned for the same test procedures 2-years after baseline. 

FTT was demonstrated to have a high intrarater and interrater reproducibility with a test-retest MDC of 15%. PwMS had significantly slower FTT time and O’Connor dexterity time when compared to pwMig and HCs and higher Fitts difficulty levels (4th and 6th difficulty) and average performance on O’Connor test were able to differentiate newly-diagnosed pwMS from HCs with 80% accuracy (p<0.01). Slower FTT performance was correlated with worse PROs due to physical health. After a 2-year follow-up, despite being clinically stable, 6 out of 8 pwMS had more than 15% worsening in their average FTT time. Therefore the Authors conclude that FTT represents a highly reproducible test for measuring psychomotor performance in newly-diagnosed pwMS and can capture insidious worsening in psychomotor performance and cognitive function in early stages of the disease.

In my opinion, this is a well executed and novel study and adds to our knowledge about the tests available for measuring psychomotor performance in the early stages of MS. The limitations are recognized and discussed.

There are few English and syntax errors throughout that could be corrected.

Response: We thank the Reviewer for the positive comments. We have further corrected the manuscript and improved on grammatical and semantical errors throughout the manuscript.

Reviewer 3 Report

1. The reviewer believes that the best structure is “‘people’ with multiple sclerosis.” The same structure should be revised for the other conditions reported in the manuscript.

Doogan C, Playford ED. Supporting work for people with multiple sclerosis. Mult Scler. 2014 May;20(6):646-50. doi: 10.1177/1352458514523499. Epub 2014 Feb 13. PMID: 24526662.

Ford HL, Gerry E, Johnson MH, Tennant A. Health status and quality of life of people with multiple sclerosis. Disabil Rehabil. 2001 Aug 15;23(12):516-21. doi: 10.1080/09638280010022090. PMID: 11432648.

2. Abstract. The last sentence of the introduction should contain the aim of the study.

3. Please, provide the IRB number.

4. Revise the abbreviation of ‘ms’ as ‘MS.’ E.g., L176

5. Was permission requested to use the “36-Item Short Form Survey?”

6. The best description for ‘education’ is ‘formal years of education.’

7. The authors should provide evidence in the discussion that ‘EDSS ≤2.0’ can significantly present subclinical disease activity. It is advised to provide results of other tests already performed in a similar population.

Author Response

Response: We thank the Reviewer for providing excellent comments and suggestions that have significantly improved the quality and readability of our manuscript. Point-by-point answers to all comments are shown hereafter.

  1. The reviewer believes that the best structure is “‘people’ with multiple sclerosis.” The same structure should be revised for the other conditions reported in the manuscript.

Doogan C, Playford ED. Supporting work for people with multiple sclerosis. Mult Scler. 2014 May;20(6):646-50. doi: 10.1177/1352458514523499. Epub 2014 Feb 13. PMID: 24526662.

Ford HL, Gerry E, Johnson MH, Tennant A. Health status and quality of life of people with multiple sclerosis. Disabil Rehabil. 2001 Aug 15;23(12):516-21. doi: 10.1080/09638280010022090. PMID: 11432648.

Response: We thank the Reviewer. There have been various usage of the term that change from one to another manuscript in the literature. We have corrected all “persons” to “people” throughout the manuscript.

  1. Abstract. The last sentence of the introduction should contain the aim of the study.

Response: This has been added in the Abstract.

  1. Please, provide the IRB number.

Response: This has been provided as requested.

  1. Revise the abbreviation of ‘ms’ as ‘MS.’ E.g., L176

Response: This has been expanded as requested. This was milliseconds and not the disease.

  1. Was permission requested to use the “36-Item Short Form Survey?”

Response: The permission for the use of 36-Item Short Form Survey has been requested by the University of Zagreb and one of the authors in this manuscript was part of the Croatian validation of SF-36. This was included in the manuscript.

  1. The best description for ‘education’ is ‘formal years of education.’

Response: This has been changed as suggested.

  1. The authors should provide evidence in the discussion that ‘EDSS ≤2.0’ can significantly present subclinical disease activity. It is advised to provide results of other tests already performed in a similar population.

Response: We have improved our Discussion and incorporated new paragraph regarding subclinical disease activity in pwMS.